# Aboriginal Young People’s Experiences of Accessibility in Mental Health Services in Two Regions of New South Wales, Australia

**DOI:** 10.3390/ijerph20031730

**Published:** 2023-01-18

**Authors:** Jasper Garay, Anna Williamson, Christian Young, Janice Nixon, Mandy Cutmore, Simone Sherriff, Natalie Smith, Kym Slater, Michelle Dickson

**Affiliations:** 1School of Public Health, University of Sydney, Camperdown, Sydney, NSW 2006, Australia; 2The Sax Institute, Glebe, Sydney, NSW 2037, Australia; 3Riverina Medical and Dental Aboriginal Corporation, 271 Edward St, Wagga Wagga, NSW 2650, Australia; 4Tharawal Aboriginal Corporation, 187 Riverside Dr, Airds, NSW 2560, Australia

**Keywords:** Aboriginal, mental health, young people, social and emotional wellbeing, cultural safety, yarning

## Abstract

This article assesses the accessibility of mainstream mental health services (MMHSs) in two regions of New South Wales (NSW), Australia, based on experiences and perspectives of Aboriginal young people aged 16–25. Semi-structured yarning interviews were conducted with thirteen Aboriginal young people in two regions of NSW. Thematic analysis was undertaken by all research team members to identify major themes from the data and conceptual connections between them. The identified themes from individual analysis and coding were triangulated during several analysis meetings to finalise the key themes and findings. Aboriginal young people had no experience of engaging with early-intervention MMHSs. MMHSs were identified as inaccessible, with most participants unaware that MMHSs existed in each region. Due to MMHSs being inaccessible, many Aboriginal young people presented to emergency departments (EDs) during a crisis. Aboriginal Community Controlled Health Services (ACCHSs) were identified as key providers of accessible, culturally meaningful, and effective social and emotional wellbeing (SEWB) service support for Aboriginal young people in NSW. If health and wellbeing outcomes are to improve for Aboriginal young people in NSW, MMHSs must increase accessibility for Aboriginal young people requiring SEWB support.

## 1. Introduction

Aboriginal young people must be able to effectively access mainstream mental health services (MMHSs) to gain support that assists them to live with and maintain positive connections to social and emotional wellbeing (SEWB). Serving as a culturally specific concept of physical and mental health for Aboriginal people, SEWB is defined as ‘a holistic concept which results from a network of relationships between individuals, family, kin and community. It also recognises the importance of connection to land, culture, spirituality and ancestry, and how these affect the individual’ [1]. Since the Ways Forward report was published, reducing the prevalence of Aboriginal young people experiencing poor SEWB has been prioritised, yet many barriers impede accessibility to MMHSs, and Aboriginal young people in Australia continue to experience complex SEWB challenges [2,3,4]. Challenges associated with accessibility of MMHSs for Aboriginal young people include lacking awareness of available services, culturally unsafe approaches to care, stigma attached to accessing support, and ineffective service provider and client relationships [5,6,7,8]. For all Australian young people, evidence has identified issues of accessibility within mainstream mental health service (MMHS) systems, including insufficient funding leading to inconsistent service availability, insufficient early intervention services with focus remaining on acute care, and difficulties associated with navigating fragmented systems that can make accessing support highly challenging [9,10].

Through existing research, we now understand quite well what living with positive SEWB means, how this may be achieved, and what this involves [11,12,13,14]; however, minimal evidence exists that identifies what restricts Aboriginal young people in NSW from accessing MMHSs to gain SEWB support [15,16]. At the same time, evidence providing insights into what happens when Aboriginal young people living in NSW do not access MMHSs when experiencing SEWB challenges does exist. Research conducted in partnership with Aboriginal Community Controlled Health Services (ACCHSs) and the Study of Environment of Aboriginal Child Resilience (SEARCH) identified high rates of suicide and presentations to emergency departments (EDs) for SEWB support amongst Aboriginal young people in communities of NSW [17], findings that reinforce the need to ensure MMHSs are more accessible for Aboriginal young people.

This article explores Aboriginal young people’s experiences and perspectives of the accessibility of MMHSs in one metropolitan and one inner-regional region of NSW. Early intervention services, MMHSs, EDs, and ACCHSs are assessed from the position of Aboriginal young people. Opportunities for reforms proposed by Aboriginal young people involved in this study are also reviewed. Through leveraging qualitative data to gain insights into issues of accessibility of MMHSs in two regions of NSW, this article provides evidence that has the potential to inform changes in NSW MMHS systems that will ensure Aboriginal young people’s SEWB support needs are more effectively addressed by increasing accessibility to MMHSs.

## 2. Materials and Methods

This article presents key findings from phase II of the SEARCH study focused on mental health and SEWB [18]. SEARCH is a longitudinal study that works in partnership with ACCHSs across NSW to improve health and wellbeing outcomes by generating data in community-nominated priority areas. Together, SEARCH and ACCHSs work to use these data to inform how health policy and service reforms can be more effective for Aboriginal communities [19]. In 2018–2020, during phase II of the SEARCH study, Aboriginal and non-Indigenous researchers worked with ACCHS staff and Aboriginal community members at one metropolitan and one inner-regional Aboriginal Community Controlled Health Services (ACCHS) to better understand what Aboriginal young people experienced when engaging with MMHS systems. ‘Aboriginal’ is used to describe the participants intentionally and with respect in this article, as all the participants self-identified as Aboriginal. Thirteen Aboriginal young people in NSW participated in yarning interviews [20,21] that explored five key themes: early intervention, access, cultural safety, service integration, and effectiveness. Deidentified data included in this article represent multiple participants from both regions. This is the first of the two articles reporting on the data from the SEWB component of phase II of the SEARCH study; this article focuses on the theme of access.

Our research team used yarning as a qualitative method for data collection. Yarning is a well-known Aboriginal research method that provides a culturally appropriate framework for engaging in discussions with Aboriginal people [20,21]. When yarning with each Aboriginal young person, we used an interview guide to create a semi-structured, in-depth interview approach that ensured all five key themes were explored. The yarning interview guide was developed by reviewing areas of concern within SEWB literature [1,16,17,22,23] and through input from the research team and the ACCHSs staff. Both Aboriginal and non-Indigenous researchers were involved in data collection, ensuring that at least one Aboriginal interviewer was always present. Gendered representation was always upheld, with all the participants offered the presence of male and/or female researchers during their yarning interview.

Aboriginal young people involved in this study were recruited with the assistance of Aboriginal health workers (AHWs) at each ACCHS. Purposive sampling was used during recruitment. All the participants were currently accessing a mental health or SEWB service or had done so within the previous 12 months. All the participants completed a Kessler 10 (K-10) psychological distress scale screening with a member of the research team prior to being recruited [24]. The participants were not eligible for yarning interviews if their K-10 scores indicated very high (>30) current levels of psychological distress or were considered by the ACCHSs staff to be too unwell to participate. All the in-depth yarning interviews were conducted on site at the participant’s ACCHS. All the participants were offered access to follow-up SEWB support from AHWs and staff before and after each yarning interview. Aboriginal young people involved in this study were aged 16–25 and received a $50 gift voucher to compensate them for their time.

Thematic analysis was used to identify major themes emerging from the data and explore conceptual connections between them. The lead author analysed each transcript to individually code important findings and develop a thematic schema. All other researchers undertook individual analysis and coding before partaking in several group analysis meetings to triangulate all identified emerging themes. In 2020, all findings were reported to each Chief Executive and key AHWs at both ACCHSs for approval.

Ethics approval for this research was obtained under the ‘Community-driven approaches to mental health service system improvements for Aboriginal children and young people’ application granted by the Human Research Ethics Committee, South Western Sydney Local Health District, NSW Health (local project No. HE18/173, HREC reference No. HREC/18/LPOOL/275). Additional approvals from the Aboriginal Health and Medical Research Council and the participating ACCHSs resulting from this application were provided.

## 3. Results

### 3.1. Theme 1: Access to Early Intervention Services

Aboriginal young people involved in this study identified a notable absence of early intervention MMHSs in each region. Although the participants had no experience of benefiting from early intervention services, they were explained by the participants when yarning as important forms of support that assist living with positive SEWB. Early intervention mental health services for young people serve crucial purposes within mental health systems through enabling early detection of mental health and SEWB challenges before more serious impacts occur, facilitating pathways for those in need of clinical support to access services, and destigmatising living with poor mental health and SEWB [25]. For Aboriginal young people, early intervention services provide meaningful opportunities for Aboriginal young people to receive education on the importance of maintaining positive SEWB and how negative SEWB challenges can be managed and mitigated [26]. By engaging with early intervention services, fundamental information about SEWB can be attained, knowledge about positive SEWB practices can be increased, and referrals to other services can be suggested when necessary [25,27]. It is crucial that Aboriginal young people can engage with early intervention supports if the process of seeking SEWB support from MMHSs is to be viewed as accessible and beneficial [28]. Yarning about the lack of early intervention services available, the participants explained how this impacted on Aboriginal young people who may have required support from MMHSs yet had no experience of doing so. Without opportunities through early intervention services to acknowledge that an individual may be experiencing SEWB challenges and require support, the participants explained that for many Aboriginal young people, it can appear that no MMHS supports exist in the local region. The participants reflected on the need to better promote that MMHSs do exist and are accessible. Targeted educational information was proposed as integral to helping Aboriginal young people understand what types of support services offer, how they differ from each other, and what may be the most suitable service based on individual circumstances [27,28].

‘I just think getting the support in there before it gets to that stage and letting them know that there is someone there. I guess they probably feel like they don’t have anybody out there. I feel like it’d be good if it was more clear to people that there is that help there.’

‘Just not knowing what you’re in for, I guess. Like just not knowing what’s going to happen and what could happen. It’s more about like not wanting to talk. I’m—a lot of people don’t realise that you got to talk about it because you’re not going to get nowhere if you don’t.’

### 3.2. Theme 2: Access to Mainstream Mental Health Services

Experiences of accessing MMHSs were predominately negative. The participants had limited awareness of MMHSs that existed in each region, and many participants explained how difficult it was to access these services and know what support is offered. The participants expressed that MMHSs were too focused on catering to those experiencing crisis situations, rather than supporting Aboriginal young people at an earlier stage of intervention. Concerning examples contextualised just how inaccessible MMHSs appear to be from the position of Aboriginal young people in this study. Feeling the need to engage in criminal activities, waiting until SEWB challenges enter a crisis phase, or acting out of character to receive attention were all common themes identified by the participants and perceived as necessary if SEWB support from MMHSs was to become accessible. The participants reflected on times when being arrested to receive a referral for support through the justice system or exhibiting extreme behaviours when presenting to MMHSs became logical pathways to access MMHSs. Prioritisation of access to MMHSs for specific Aboriginal young people was explained as an inequitable and exclusive approach.

‘It’s—the only way I think you can get help is if you go absolutely crazy and the police are called in, you’re shipped in there, or something. It’s the only way you’re going to get help. I didn’t want to get to that stage, but it was crossing my mind, maybe I have to do something stupid to get help. Because no one would give me help. I tried every phone number, every avenue, emergency, every mental health facility. Turned away. Yeah.’

‘Well, there’s no supports really available unless you go get locked up or go get a criminal activity. That’s the only support that you have, is through juvenile justice. They’re the only people that can support anyone in my eyes…’

‘Yep, because up here, there’s not many programs for younger than my age, and I’m 24. Name one program I’ve come to here this week, any day. I’ll have a day off it. There’s not one program I can come, and I don’t have children, but why should I have to have children to come to a program?…Why should I have to go through Domestic Violence to come to a [SEWB] program?’

### 3.3. Theme 3: Access to Emergency Departments

Lacking access to early intervention and MMHS supports, presenting to EDs for SEWB support was a common experience amongst the participants. Aboriginal young people felt they had to wait until SEWB challenges placed them in a crisis to present to EDs, where multiple negative experiences occurred. Inadequate mental health assessments were consistently raised as key barriers in EDs. The participants who had reached a crisis point were usually not offered a mental health assessment or attended to by mental health specialists. On rare occasions when the participants were assessed by mental health specialists at EDs, the participants reported that the assessments were inadequately conducted, and the SEWB problems explained to the mental health specialists were not taken seriously. Examples provided by the participants included being given antidepressant medication and encouraged to go home and relax, given a suicide hotline number that was meant to be contact information for a local MMHS to arrange appointments, having no follow-up communication about future support options after eventually completing assessments with ED staff, and being unable to access ED mental health supports during a severe panic attack.

‘Positive? None. None at all, to be honest. Negative, it’s all negative, really. I couldn’t—in the emergency department, you do not get to see someone from mental health. Normally, there’s one—there’s at least someone there from mental health. Do not even get close to seeing them. I also have a partner who was admitted, same kind of thing, mental health. He had a severe panic attack. He didn’t even get to touch base with mental health, and he’s not even Aboriginal. So, if they can’t access it, how are we going to access it? It’s near impossible. You’ve pretty much got to be, I think, on a court order or something to get the service.’

### 3.4. Theme 4: Access to Aboriginal Community Controlled Health Services

Aboriginal Community Controlled Health Services (ACCHSs) were the first point of contact and main providers of SEWB support for Aboriginal young people, with experiences being majorly positive. ACCHSs provide health and SEWB services to Aboriginal communities through culturally informed and unique models of care, operating through culturally safe and holistic approaches, usually provided by AHWs, that are underpinned by the understanding of the Aboriginal community and client support needs [29,30]. ACCHSs do operate differently from MMHSs and EDs, which positions Aboriginal clients with the need to utilise EDs and MMHS for certain services and supports, including emergency and afterhours access, and when requiring forms of more acute mental health and SEWB clinical care. With ACCHS staff members often part of the local Aboriginal community, acting as key stakeholders in improving health and SEWB outcomes of the local community, most Aboriginal young people felt more confident in accessing SEWB support through local ACCHSs. The participants explained that by knowing they would engage with Aboriginal staff, their SEWB challenges would be taken seriously, understood, and respected. Yarning further about the importance of ACCHS SEWB supports and the positive impact of having Aboriginal staff within mental health services, the participants justified that by having SEWB support from staff who knew the reality of what was going on in the local community context, the support needs of Aboriginal young people did not seem unrealistic or unreasonable. Therefore, the benefits of accessing services were not doubted by Aboriginal young people in need of SEWB support [30,31]. ACCHS provisions of SEWB outreach services, flexible appointment times, and person-centred care prioritising meaningful relationships between a client, staff, and clinicians were all unique factors that strongly supported Aboriginal young people to access ACCHS SEWB supports.

‘Yeah, you can just call up and the girls straight away direct you to where you need to go. Yeah, you can book an appointment. For a crisis, like I said before, you can just drop in or they can come out to you. They’re quite flexible, yeah.’

‘I think out here is really good, how—especially, like I was saying before, probably multiple times that you guys can actually outreach and come out to people. Whereas if they’re having anxiety and they really can’t bring themselves to come in here, you guys are willing to come out to them, which is really good.’

### 3.5. Theme 5: Suggestions from Aboriginal Young People

All the participants in this study proposed reforms that have potential to enhance the accessibility of MMHSs for Aboriginal young people. Lacking experience of benefiting from early intervention services, being burdened by accessibility challenges within MMHS systems, and with predominately negative MMHS experiences all common themes amongst the participants, the suggested reforms targeted better understanding the needs of Aboriginal young people accessing service support, increasing attention towards the importance of person-centred care, and the provision of relevant information which could support the help-seeking process when accessing MMHSs.

Increasing the availability and quality of early intervention services was deemed a priority by Aboriginal young people. The participants wanted early intervention services to target other Aboriginal young people from a much earlier age, ideally beginning in the early primary school years. The participants suggested that early intervention services should be holistic, involving culture, SEWB, physical activity, and education that is inclusive of the broader community’s SEWB needs. Building better partnerships between the key stakeholders in the Aboriginal young people’s health and SEWB was a practical suggestion raised on multiple occasions. ACCHSs, Aboriginal liaison officers (ALOs), AHWs, schools, and MMHSs were challenged to work more closely together to implement meaningful and relevant SEWB early intervention services for Aboriginal young people.

Regarding service and client relationships, the Aboriginal young people desired to be treated with greater respect, for SEWB challenges to be taken seriously, and to enhance person-centred care. Taking time to understand individual, community, and cultural factors impacting on SEWB was integral to improve accessibility and retain Aboriginal young people as clients. For many Aboriginal young people, accessing an MMHS was explained as daunting. The participants expressed the need to allocate more time to building meaningful relationships with Aboriginal young people to ensure they feel comfortable accessing MMHSs.

Targeted health promotion advertising of SEWB and mental health content was another common suggestion that could enhance accessibility. The participants felt that to increase the number of Aboriginal young people accessing MMHSs, they fundamentally required information about what is on offer, who is involved, how this helps them, and why this is important. Aboriginal young people explained that advertisement should utilise technology and be delivered in ways that are engaging for Aboriginal young people based on technological usage within the community context.

In terms of MMHSs and EDs, Aboriginal young people wanted mental health assessments to be conducted properly, information on MMHSs to be accurate, and, where possible, greater flexibility for scheduling and attending appointments. Many Aboriginal young people that had eventually made the difficult yet important choice to seek SEWB support from an MMHS or presented to EDs were provided with inaccurate information about a service or were not assessed properly. On numerous occasions when yarning, the participants explained how challenging it can be for an Aboriginal young person to seek support from an MMHS to then be turned away or not gain the support that was much needed. Multiple participants explored personal stories or those of friends in the local community who had had these negative experiences and had never attempted to access MMHSs again. Proposed reforms for MMHSs included remaining flexible for Aboriginal young people who may experience challenges making appointments, providing follow-up calls before and after appointments, and by providing information for clients to take home that supports maintaining positive SEWB between appointments.

‘Just definitely a more supports-based service, and if you’re going to be accessing [SEWB] through mainstream, the mainstream needs a whole big overhaul especially. Because you can’t access it. It’s all for show…It definitely does need extra support, especially if it’s out of the grounds of here [ACCHS]. I think if it’s more in a mainstream environment where you’re getting assistance from, I think there needs to be more support, especially for Aboriginal people, but also the normal community as well. Because it seems like nobody can really access it. It’s all just for show, practically. Yeah.’

‘I think talking with the person and explaining things to them nice and clearly. Because, I know with anxiety you can get quite worked up, and it just goes through one ear and out the other. I think just having the comfort zone and knowing that you’re not going to be turned away. My attitude going into emergency was shocking because I knew it was just going to be the same thing. Here’s a Valium, out the door you go for the night. That’s that. It doesn’t solve the problem. There’s obviously an underlying issue, and it never gets delved into. Yeah.’

‘Advertising would help. Kids are all into technology these days so advertisement on phones or in their schools or even if parents talk about it at home. I know growing up that no one ever talked about that in their household.’

‘If you’re looking at younger kids at school, like going to the Aboriginal liaison officer at school with your thoughts or anything like that could help, and then, I guess, they can refer to seeing the doctor on going on from there, going to headspace; things like that.’

## 4. Discussion

The Aboriginal young people involved in this study considered MMHSs to be largely inaccessible. The participants clearly identified that many Aboriginal young people in NSW requiring SEWB support are not able to access the help they need. Beginning with early intervention, at the primary services level, and through to tertiary care, issues of inaccessibility were identified at every stage of the care continuum. Highlighting the extent of inaccessibility was the participants’ consistent reflection on the crisis-driven nature of the MMHS system. For the common lived experience of Aboriginal young people to have no engagement with early intervention supports, to be unaware of MMHSs available in the local region, and presenting to EDs in crisis situations, MMHS systems can be viewed as currently failing to provide the support they are designed to offer for Aboriginal young people. Further, when considering the National Mental Health Commission’s view that individuals presenting to EDs seeking mental health support indicates the failure of MMHS systems at earlier stages across the care continuum [17,32], barriers to gaining SEWB support due to inaccessibility become urgent challenges that must be addressed.

Increasing the availability and enhancing the effectiveness of early intervention services targeting Aboriginal young people’s SEWB must be prioritised [3]. Most participants involved in this study had no awareness or experience of early intervention services that existed in the local region. Without early intervention services providing opportunities for Aboriginal young people to receive informal support before more serious SEWB problems develop, education on the importance of maintaining positive SEWB, being informed about what types of SEWB support exist in the local MMHS system, variations between services and differing purposes, and how these can be accessed, the likelihood of accessing MMHSs is significantly decreased.

The participants reinforced the importance of providing early intervention services for all young people, not just for those with specific needs or temporary circumstances warranting unique forms of access to support [6,33]. Ideally, early intervention services should be introduced to children in primary school and remain consistently available up until twenty-five years of age [3]. The participants shared that if early intervention supports are to be implemented effectively, they should function through the locally informed understanding of culture, health, and SEWB, have holistic approaches to engaging participants in activities that help to explore the importance of positive SEWB, how to manage SEWB challenges, and serve as an integration pathway to other MMHS supports that may be required [34]. Ideally, early intervention supports should be provided by ACCHSs, as these services are already known to be leaders in providing accessible and culturally meaningful SEWB supports for Aboriginal young people. For this to occur, ACCHSs must be financially supported to develop early intervention services and employ more AHWs to facilitate them. MMHS systems must also increase financial, staffing, and strategic investments into embedding accessible, culturally meaningful, and effective early intervention services and support for Aboriginal young people, ensuring that the responsibility to support Aboriginal young people’s SEWB does not solely rely on ACCHS services and supports.

Mainstream mental health services (MMHSs) must do better to increase the accessibility and availability of support for Aboriginal young people. Too many participants believed that the only way to receive support from MMHSs was to be in a crisis or dangerous situation. If an Aboriginal young person is to experience being denied access to multiple services within the local MMHS system, not taken seriously when presenting at the service, or not assisted to arrange support at a future time or from another suitable service, seeking support will likely not continue [7,35]. Allocating sufficient time to build meaningful relationships with Aboriginal young people as clients was one identified reform that should be prioritised [36]. Explaining how MMHSs function, what Aboriginal young people can expect from the service in terms of client support, and the provision of accurate mental health support information that Aboriginal young people can utilise in between appointments are all relatively simple reforms that would be meaningful for Aboriginal young people as clients of MMHSs.

For MMHSs that cater to young people yet do not currently support Aboriginal young people, building more supportive relationships with local Aboriginal communities and ACCHSs should be prioritised to assess how this can be best approached. MMHSs should be responsive not only to individual Aboriginal young people presenting for support. Rather, ACCHSs, AHWs, ALOs, and community leaders should be engaged with to help encourage access of MMHS supports by Aboriginal young people that is available yet not being utilised. To further increase accessibility, targeted mental health promotion content explaining how to access SEWB support available in the MMHS system should be promoted through technological avenues relevant to Aboriginal young people and be inclusive of as many MMHSs as possible.

Although EDs are not specifically designed to support mental health and SEWB, most Aboriginal young people who had presented to EDs in a crisis were unable to access support from mental health staff. Of concern was the Aboriginal young people sharing broader community beliefs that EDs were the only place you could genuinely have some chance of gaining SEWB support. If Aboriginal young people are unable to gain support from MMHSs and present to EDs, accessibility to ED mental health staff and supports must be enhanced through conducting proper mental health assessments. Mental health staff, AHWs, and ALOs must be able to offer mental health assessments for Aboriginal young people in ED settings. If mental health assessment outcomes lead to an Aboriginal young person being encouraged to leave the ED and seek support elsewhere, it is crucial that every person has a follow-up support plan. Communicating where further MMHS support exists, how this can be accessed, and who can help with making these arrangements are all fundamental actions that should be common practice. AHWs and ALOs should be present during this communication process, with local ACCHSs being made aware of what the Aboriginal young person has explained about their SEWB experiences and support needs.

For MMHSs in NSW to be more accessible for Aboriginal young people, ACCHSs should be consulted about what works best and how this can be adapted into existing services. All the participants in this study had positive experiences when reflecting on the forms of SEWB support that ACCHSs offer Aboriginal young people in the local community. SEWB service support that is grounded in culturally informed approaches to care, guided by understanding SEWB challenges that the Aboriginal community experiences, and factors in the need for service flexibility for Aboriginal young people are features of a unique model of SEWB support that must be considered by MMHSs supporting Aboriginal young people.

## 5. Conclusions

From the position of Aboriginal young people in this study, MMHSs have severe accessibility issues across all stages of the care continuum. Meaningful reforms that have potential to improve the accessibility of MMHSs for Aboriginal young people were identified by the participants. Culturally meaningful SEWB early intervention services should be implemented. MMHSs must improve accessibility by providing accurate information that helps Aboriginal young people to understand how to access support by increasing attention towards providing person-centred care and through developing a better understanding of what support Aboriginal young people require when navigating MMHS systems to access SEWB support. ED staff must ensure mental health assessments are conducted properly and that Aboriginal young people are successfully supported to seek SEWB support from the MMHSs available within the local region. ACCHSs should be recognised as providers of meaningful SEWB service supports and serve as highly valuable contacts for MMHSs attempting to implement reforms to enhance accessibility for Aboriginal young people. If health and wellbeing outcomes are to improve for Aboriginal young people in NSW, MMHSs must increase accessibility for Aboriginal young people requiring SEWB support.

## Data Availability

The data are not publicly available from this research due to the personal and sensitive nature of the participants sharing qualitative lived experience of mental health service experiences.

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
