# Peer review of "Aboriginal Young People’s Experiences of Accessibility in Mental Health Services in Two Regions of New South Wales, Australia"

_ijerph, 2023, doi:10.3390/ijerph20031730_

Round 1
Reviewer 1 Report
Comments and Suggestions for Authors This is a critically important, well written and well designed research study for the serious issue of mental health and wellbeing in Aboriginal youth. It should be published. My suggestions for improvement are: 1. More detail is provided on what is meant by "early intervention" and the evidence base for this. 2. Recommendations should include a greater depth of discussion regarding grounded strategies to provide early intervention - ie Should this be located at the Aboriginal Community Controlled Health Services (ACCHSs), identified as key providers of accessible, culturally meaningful care? What should it look like, based on the evidence and the participants' suggestions? 3. Recommendations should address the critical staff shortages in mental health - are there any alternative pathways ie training AHWs in mental health assessment and early intervention, providing generic staff with time for liaison with ACCHSs etc, more proactive recruitment to Mental health positions for Aboriginal mental health workers? 4. Consideration that the target audience of this journal is international and therefore may not have the full context of the determinants of poor SEWB in Aboriginal Australians, nor what ACCHs offer (ie do not offer emergency services/ED. Brief additions to address these should be included.Author Response
Thank you for reviewing this article and providing useful, specific feedback. I have addressed the below points by editing and addressing the following:
1. More detail is provided on what is meant by "early intervention" and the evidence base for this: Have added up front explanation of what early intervention services involve and important role in mental health service systems - lines 126 - 130.
2. Recommendations should include a greater depth of discussion regarding grounded strategies to provide early intervention - ie Should this be located at the Aboriginal Community Controlled Health Services (ACCHSs), identified as key providers of accessible, culturally meaningful care? What should it look like, based on the evidence and the participants' suggestions: Have added depth to this response, stating this should ideally occur through ACCHSs, however this requires financial and staffing support, and that mainstream mental health service systems should also share this responsibility sand strategic investment. Have also justfified what these services should look like based on particpant data - lines 336 - 351.
3. Recommendations should address the critical staff shortages in mental health - are there any alternative pathways ie training AHWs in mental health assessment and early intervention, providing generic staff with time for liaison with ACCHSs etc, more proactive recruitment to Mental health positions for Aboriginal mental health workers: As this is paper 1 of 2, this information was specifically included in the 2nd article working with the same data that focuses on cultural safety. Staffing across the whole system, the role of ACCHSs and AHWs, and how these can be improved are all discussed. This was an intentional decision made by the authoring team.
4. Consideration that the target audience of this journal is international and therefore may not have the full context of the determinants of poor SEWB in Aboriginal Australians, nor what ACCHs offer (ie do not offer emergency services/ED. Brief additions to address these should be included:
Our author team did not want to spend too much time dwelling on the SEWB inequities and challenges experienced by Aboriginal young people, and instead, really delve into the service and systems experiences and accessibility issues, hence the limited discussion on determinants of SEWB. I have added a sentence to reinforce that SEWB challenges remain common and complex - line 40.
After reviewing lines 216 - 222, I think these role and functions of ACCHSs have been addressed sufficiently, however have taken on the advice and added they do not directly deal with hospital emergency departments/services and operate differently in some aspects to EDs and mainstream services - lines 219 - 222.
Thank you for providing such detailed feedback! This is my first ever academic publication as lead author/any author so I'm feeling very excited to be so close to publication.
Much appreciated,
JG
Reviewer 2 Report
there are participant quotes, but no one is ascribed to these. It would be good to know if they are different/same participant.
Discussion: participants haven't accessed/didn't know about early intervention services, but advocated for them. Not clear if this was a question that was asked or it came about through yarning.
Great paper overall and provides much-needed content for Aboriginal youth and mental health.
Author Response
There are participant quotes, but no one is ascribed to these. It would be good to know if they are different/same participant:
Have added a line to explain multiple participants are represented through data in both regions, however will remain de-identified - line 83 - 84.
Discussion: participants haven't accessed/didn't know about early intervention services, but advocated for them. Not clear if this was a question that was asked or it came about through yarning.
Have clarified this wording to ensure readers know this was stated by participants when Yarning.
Thank you for reviewing my first ever lead author publication!